

# Community composition and seasonal changes of archaea in coarse and fine air particulate matter

Jörn Wehking[1,2], Daniel A. Pickersgill[1,2], Robert M. Bowers[3,4], David Teschner[1,2], Ulrich Pöschl[2], Janine Fröhlich-Nowoisky [2], Viviane R. Després[1,2]

[1]Institute of molecular Physiology, Johannes Gutenberg University, Johannes-von-Müller-Weg 6, D-55128 Mainz, Germany

[2]Max Planck Institute for Chemistry, P.O. Box 3060, D-55020 Mainz, Germany

[3]DOE Joint Genome Institute, Walnut Creek, CA, USA

[4]University of Colorado, Boulder, CO, USA.

*Correspondence to*:

Viviane R. Després (despres@uni-mainz.de)

Jörn Wehking (wehking@uni-mainz.de)

**Abstract.** Archaea are ubiquitous in terrestrial and marine environments and play an important role in biogeochemical cycles. Although air acts as the primary medium for their dispersal among different habitats, their diversity and abundance is not well characterized. The main reasons for this lack of insight is that archaea are difficult to culture, seem to be low in number in the atmosphere, and have so far been difficult to detect even with molecular genetic approaches. However, to better understand the transport, residence time, and living conditions of microorganisms in the atmosphere as well as their effects on the atmosphere and vice versa, it is essential to study all groups of bioaerosols. Here we present an in-depth analysis of airborne archaea based on Illumina sequencing of 16S-rRNA from atmospheric coarse and fine particulate matter samples and show seasonal dynamics and discuss anthropogenic influences on the diversity, composition, and abundance of airborne archaea.

The relative proportions of archaea to bacteria, the differences of the community composition in fine and coarse particulate matter, as well as the high abundance in coarse matter of one typical soil related family, the Nitrososphaeraceae, points to local phyllosphere and soil habitats as primary emission sources of airborne archaea.



We found comparable seasonal dynamics for the dominating Euryarchaeota classes and

Crenarchaeota orders peaking in summer and fall. In contrast, the omnipresent Cenarchaeales

and the Thermoplasmata occur only throughout summer and fall. We also gained novel insights

into archaeal compositon in fine particulate matter (<3 µm), with Cenarchaeaceae,

Nitrososphaeraceae, Methanosarcinales, Thermoplasmata and the genus *Nitrosopumilus* as the

dominating taxa.

The seasonal dynamics of methanogenic Euryarchaeota points to anthropogenic activities, like

fertilization of agricultural fields with biogas substrates or manure, as sources of airborne

archaea. This study gains a deeper insight into the abundance and composition of archaea in the

atmosphere, especially within the fine particle mode, which adds to a better understanding of the

overall atmospheric microbiome.

**1    Introduction**

Besides bacteria and eukaryotes, archaea are regarded as a third independent domain of life

(Woese et al., 1990). In the beginning of archaeal research in the 1880s primarily methanogenic

archaea were discovered and cultivated, so the belief arose that archaea are exclusively

extremophiles (Cavicchioli, 2011; Farlow, 1880; Schleper et al., 2005). However, during the last

decades, cultivation and culture independent methods, like DNA sequencing approaches, have

substantially improved our understanding of archaea and proved that they are also abundant in

various environments such as marine, or soil habitats where they can represent more than 10 %

of the microbial community (Buckley et al., 1998; Cao et al., 2012; Cavicchioli, 2011; Delong,

1998; Robertson et al., 2005; Yilmaz et al., 2016).

So far, diversity studies for archaea have mainly concentrated on the major habitats known also

for bacteria such as marine and soil environments (Bintrim et al., 1997; Buckley et al., 1998;

DeLong, 1992; Ochsenreiter et al., 2003). In the global marine environment the abundance of

archaea is approximately $1 \times 10^{28}$ archaeal compared to $3 \times 10^{28}$ bacterial cells (Karner et al.,

2001) with archaea accounting for 2-10 % in surface waters and for 20-40 % in deep ocean water

(Massana et al., 1997).

The abundance and composition of archaea in soil varies between different soils types (Bates et

al., 2011). All cultivated methanogens belong to the kingdom Euryarchaeota and are strictly

depending on anaerobic conditions with low redox potentials (Le Mer and Roger, 2010), thus




they are present in many soils only in small numbers. The fertilization with life-stock manure

adds anaerobic adapted organisms to the surface of agricultural used soils. Thus, even in aerated

soils, core anaerobic populations seem to survive though in low number (Angel et al., 2012).

Another issue influencing the abundance and composition of archaea in soil is - as also observed

in water columns- the depth (Karner et al., 2001). Analyses of soil depth profiles revealed

changing diversity patterns with depth (Bundt et al., 2001; Pesaro and Widmer, 2002) in

composition and number.

Next to the well-established major habitats, the atmosphere is another environment in which

microorganisms are detected, though it remains unclear whether the atmosphere can be

considered a natural habitat or represents only a medium of dispersal for terrestrial and marine

microorganisms and their spores (Bowers et al., 2009, 2011, 2012, 2013; Womack et al., 2010;

Yooseph et al., 2013). For airborne bacteria and archaea the main known emission sources are

surface waters and the surface layer of soils (Womack et al., 2010). Therefore, the different

abundances and composition of archaea within water and soil columns are of special interest to

understand possible emission sources for airborne archaea. For bacteria, which are abundant in

air, the concentration of DNA 16S rRNA gene copies (cp) quantified using qPCR in soil was

$10^{11}$ to $10^{12}$ DNA cp $kg^{-1}$ and $10^{9}$ to $10^{11}$ cp $kg^{-1}$ for archaea (Cao et al., 2012; Kemnitz et al.,

2007). In ocean surface waters the concentration is lower but estimated to be $10^{8}$ to $10^{9}$ cp $L^{-1}$ for

bacteria and $10^{6}$ to $10^{7}$ cp $L^{-1}$ for archaea (Kemnitz et al., 2007; Yin et al., 2013) whereas only

$10^{4}$ to $10^{6}$ bacterial cp $m^{-3}$ air were detected (Cao et al., 2012; Fröhlich-Nowoisky et al., 2014;

Kemnitz et al., 2007; Yin et al., 2013). Interestingly in contrast to bacteria, it seems challenging

to detect, amplify, and analyze archaea in air, as their concentration of 100 ppm is much lower

than the abundance of bacteria (Cao et al., 2012; Fröhlich-Nowoisky et al., 2014). Until now, it

remains unclear whether these observations are biased by technical obstacles or reflect the true

abundances. The largest study on airborne archaea is to our knowledge by Fröhlich-Nowoisky et

al., (2014) and is based on Sanger sequencing. However, in Fröhlich-Nowoisky et al., (2014) the

number of sequences were low, the observations are with little statistical support and the analysis

of the microbiome of aerosolized archaea is difficult. Therefore, we present an in-depth

pyrosequencing study of airborne archaea collected on coarse and fine particulate matter filters

over one year in Mainz, Germany. We attempt to compare the composition, diversity, and



1  abundance to the same characteristics as in other habitats, which also allows an inference about

the     primary     emission     sources     of     airborne     archaea.





## 2 Material and Methods

### 2.1 Aerosol sampling

As described in (Fröhlich-Nowoisky et al., 2009), in total 24 pairs of air filter samples (i.e., 20 filter pairs of one fine and one coarse particle filter sample each, two pairs of start-up air filter blanks and two

pairs of mounting filter blanks) were analyzed within this data set. The air filters were installed on a self-built high-volume-dichotomous sampler (Solomon et al., 1983). The whole sampling campaign lasted one year in Mainz, Germany (March 2006 - April 2007). The rotary vane pump (Becker VT 4.25) worked with a flow rate of $\sim 0.3$ m$^3$ min$^{-1}$, corresponding to a nominal cut-off diameter of $\sim 3$ µm. The particles with an aerodynamic diameter larger than $\sim 3$ µm and 10 % of the fine particles were collected

on one glass fiber filter ($\sim 0.03$ m$^3$ min$^{-1}$) representing the coarse fraction. The fine particles from the same air mass were collected on the corresponding second glass fiber filter ($\sim 0.27$ m$^3$ min$^{-1}$) which was essentially free from coarse particles (Solomon et al., 1983). Except for filter pairs MZ 11 (24 h) and MZ 15 (5 d), all filter pairs were collecting air over a 7 day period (Table S1). The sampled air masses represent a mixture of urban and rural continental air, as the sampler was positioned on the roof of the

Max Planck Institute for Chemistry on the campus of the University of Mainz (49°59'31.36''N, 8°14'15.22''E). To reduce the sampling of particles emitted from the ground, the sampling device was on a mast about 5 m above the flat roof of the three-story building.

### 2.2 Extraction, amplification, and sequencing

The DNA extraction and sequencing was part of the Earth Microbiome Project (EMP -
http://www.earthmicrobiome.org/) using the Illumina GAIIx sequencer with the pyrosequencing technology. As shown before, this technology is suitable for analyzing microbial communities in soil, water, and human skin (Caporaso et al., 2011).

For the PCR amplifications the 515f/806r primer set described in Caporaso et al., (2011) proved to be most suitable. It covers the conserved flanking regions ideal for amplifying bacteria and archaea over

the V4 region of the 16S rRNA gene (Bowers et al., 2013; Huse et al., 2008; Muyzer et al., 1993). In addition, the primer pair is preferred for this amplification as it exhibits only few biases against





individual bacterial taxa. As suggested in Caporaso et al., (2011) each DNA extract was amplified in triplicate. These triplicates were combined and purified using a 96 well PCR clean-up kit from MO BIO. The utilized PCR reaction was performed; amplicons purified and sequenced using the GAIIx.

## 2.3    Grouping of sequences into OTUs and taxonomic identification

The sequences were analyzed using the Quantitative Insight Into Microbial Ecology (QIIME) toolkit (Caporaso et al., 2010). To assign sequences to OTUs, we used Qiime's closed reference OTU picking script which uses Uclust (Edgar, 2010) and the greengenes reference database (gg_13_8_otus/rep_set/97_otus.fasta, last update 08/15/2013 ; McDonald et al., 2012) with 97 %
similarity. For the actual identification process a corresponding taxonomy map provided by the greengenes database was used. Sequences, which did not match to any greengenes reference set OTU, were discarded for the downstream analysis.

## 2.4    Controls

Prior to the sampling procedure all filters were baked in sealed aluminum foil bags overnight at 500° C.
To best conserve the DNA of the collected bioaerosols, after the sampling procedure all filter samples were stored at -80° C until analysis. To detect possible contaminants from the sampling device and also the filter handling, blank filters were taken at 4-week intervals. Contamination free, prebaked filter pairs were mounted in the sampler as for regular sampling, but the pump was not turned on at all ("mounting blanks"). In addition, small environmental samples were taken to collect air exclusively around and
from the interior of the sampling device by turning the pump on for 5 s only (start-up filter blanks). A detailed list of all analyzed air and blank filter samples with their individual sampling details can be found in the supplemental material in Tab. S1.

The DNA of the blank filters was extracted parallel to the actual filter samples and quantified. Often, the detected DNA concentrations on such blanks is too small to be quantifed or to build usable
sequencing libraries (Cao et al., 2014). However, as shortly exposed to environmental air, they also can contain DNA. Within this study we controlled the actual filter changing process by sequencing two



mounting blanks, i.e., MZ 23 und MZ 73. Two sequences were obtained from the fine particle filter of MZ 23 and 408 archaeal sequences (371 on the coarse and 37 sequences on the fine particle blank filter) were detected in the air mass of MZ 73. On the coarse particle filter of MZ 23 no archaeal sequences were detected. A minimal DNA amount here is likely, as the filters are shortly exposed to the rural-

urban continental air of the sampling site during the mounting process. The resulting sequences of the mounting blanks were analyzed alongside with the other sequences and could be assigned to five archaeal families (Cenarchaeaceae, Methanobacteriaceae, Methanoregulaceae, Methanosaetaceae, Methanomassiliicoccaceae). In the handling of the sequences obtained in next generation sequencing techniques, e.g., for amplicon sequencing of environmental air samples controls are neither well

established nor standardized. To ensure that all contaminants were removed comprehensively from the data set, we decided to omit all identified families from the data if present in more than 1% of all detected archaeal sequences of the mounting blanks.

The thus deleted families (404 sequences) belong to the Euryarchaeota and could be assigned to the following families (see also Table S2): the Methanoregulaceae (8.5%, 3 OTUs),

Methanomassiliicoccaceae (17.6%, 3 OTUs) and the largest family of the Methanobacteriaceae (72.4%, 4 OTUs). Thus, 2341 sequences remained for the downstream analysis.

The two pairs of start-up air filter blanks were sequenced likewise. But as they were sampled for five seconds the obtained sequences were not treated like the mounting blanks. On these four filter samples 709 archaeal sequences were found, distributed with 328 sequences on MZ 22 (326 sequences on

coarse, 2 sequences on fine) and 381 sequences on MZ 72 (3 sequences on coarse, 378 sequences on fine).

### 2.5    Statistical analysis

All data management and most of the analyses were performed using a MySQL database and R-Statistics if not stated otherwise (R-Team, 2011).

To characterize the biodiversity of the archaea community and thus to approximate the likely diversity several statistical parameters were calculated: species richness estimators, rarefaction curves, and community diversity indices using the software tool EstimateS (Colwell et al., 2012)



### 2.6 Meteorological analysis

As a possible correlation between the abundance of taxonomic ranks in an air mass and meteorological parameters can be either following a monotone or specifically a linear relationship, in this study the Pearson product-moment correlation coefficient ($r_K$) testing for a linear regression and the Spearman's

Rank ($r_K$) for fine, coarse and total suspended particles. The meteorological parameters tested were: wind speed in m s$^{-1}$ (average and maximum), temperature in °C (range and maximum), relative humidity in %, and the sum of precipitation in mm. The meteorological data were provided by full hour data for wind speed and half hour values for all other meteorological parameters by the ZIMEN Luft Messnetz of the Landesamt für Umwelt Wasserwirtschaft und Gewerbeaufsicht of Rhineland Palatine.

All averages were calculated for the exact sampling periods (Tab. S1). The correlation analysis using the Pearson product-moment correlation coefficient ($r_K$) and Spearmans Rank ($r_R$) was calculated on different taxonomic levels, i.e., on kingdom, phylum, and class level. Only results with $r_K$ or $r_R$ over 0.5 or under -0.5 were interpreted. As resuming information we tried to find significant correlations between the relative abundance and the meteorological factors, but did not find provable significant

correlations.

### 3 Results and Discussion

### 3.1 Overall Diversity

To determine the archaeal diversity in air, 20 air filter pairs were sampled and analyzed for one year in Mainz, Germany. Each filter pair consists of one filter collecting particles with aerodynamic diameters

smaller than 3 μm (fine particulate matter) and one collecting primarily coarse particles, which are larger than 3 μm. On 39 (97.5 %) of the 40 analyzed filters (20 air filter pairs) archaeal DNA could be detected. In total 2.341 sequences could be assigned to archaea (Tab. 1). More archaeal sequences were detected on coarse particle filters (109 sequences on average per sample) than on fine particle filters (8 sequences on average per sample) for which the number of sequences ranged from 0 to 42. On all but

one fine particle filter, MZ 81 sampled in December 2006, archaeal sequences were discovered. Most sequences, i.e., 601, were detected on the coarse particle filter MZ 74 from November 2006. These



obtained 2,341 archaeal sequences were assigned to 52 OTUs. Out of these OTUs, 17 OTUs were found in coarse as well as in fine particulate matter. As listed in Table 1 the coarse particle filters comprised 2180 sequences distributed among 41 OTUs whereas only 161 sequences assigned to 28 OTUs could be identified on the fine particle filters.

In total only 7 % of all archaeal sequences stem from fine, whereas 93 % stem from coarse particle filters. Specifically, on 75 % of the coarse particle filters 20 or more archaeal sequences were found, while on 70 % of the fine particle filters less than six archaeal sequences could be detected.

The community structures of both size fractions differ remarkably in their composition (Fig. 1). In the fine fraction the genus *Nitrosopumilus* is the dominant taxon. This Thaumarchaeota genus shows a

relative abundance of 33.5 % over all archaea sequences found on all samples in fine particulate matter. The cultivable *Nitrosopumilus maritimus* is a well-known representative of the genus *Nitrosopumilus*. These chemolithoautotrophic nitrifying archaea have been sampled primarily from marine sources. They form straight rods with a diameter of 0.17–0.22 µm and a length of 0.5–0.9 µm (Könneke et al., 2005) and are thus one of the smallest organism known today. With this size even long distance

transport from marine sources might be conceivable. The same can be said for species of Marine group II. However, *N. maritimus* and species of Marine group II have been found in soil samples (Leininger et al., 2006; Treusch et al., 2005). In contrast to the coarse fraction, where only the genera *Methanocella* and the Candidatus *Nitrososphaera* were found with a relative proportions of more than 3 %. As on coarse particle filters many more sequences could be analysed compared to the fine particle filters,

analysis of the total suspended particles (TSP) resemble the results of the coarse particles (Fig. 1). Taking the relative distribution over the entire course of the year into account, on class level the Thaumarchaeota also dominate the fine particle fraction. Except for two fall fine filters where the Euryarchaeota even have a higher relative abundance than the Thaumarchaeota (93 % and 92 %).

The Crenarchaeota, primarily determined by Thaumarchaeota (99 %), are the dominating phylum in the

coarse particle mode. Next to Thaumarchaeota the Miscellaneous Crenarchaeotal Group (MCG; Kubo et al., 2012) was exclusively found on the coarse spring filter with seven sequences representing one single OTU. No Euryarchaeota were observed on 65 % of the fine and 50 % of the coarse particle filters. On a closer look among taxonomic assignments the contribution of sequences to individual





families reveals that within the coarse particle fraction most sequences belong to the Nitrososphaeraceae family. While this family is only present in 10 % of the fine particle filters it could be identified in 75 % of the coarse particle filters. In soil surveys the I.1.b group of Crenarchaeota has constantly been found (Ochsenreiter et al., 2003) with the Nitrososphaeraceae being one of the most abundant archaea family

therein. Thus, for this family primarily the aerosolization of soil and soil dust can be hypothesized. Within this family the genus Nitrososphaera is an abundant taxon in specifically agricultural soils (Zhalnina et al., 2013). The landscape of the surrounding area of the sampling location is dominated by agricultural fields and the emitted soil particles are thus, likely to contain the genus *Nitrososphaera*. Soil and soil dust are classically discussed as primary emission sources for airborne bacteria (Després et

al., 2007, 2012; Fierer et al., 2008). Therefore, when attached to large soil particles these organisms should be mainly collected in the coarse particle fraction. To our knowledge, the only cultivated *Nitrososphaera* species, *Nitrososphaera viennensis,* has a much smaller diameter (irregular cocci with a diameter of 0.6–0.9 µm; Stieglmeier et al., 2014), which should be, if in single cell status, collected in the fine particle mode. The hypothesis that soil particles identified through Nitrososphaeraceae are

mainly collected on coarse particles is also strengthened by the results of community analysis of the fine particle filters. The observed increase of the relative abundance of the Euryarchaeota can be interpreted as the decline of Nitrososphaeraceae because soil particles are less frequent in the fine mode. So at least on phylum level the Nitrososphaeraceae family forms the main difference between the two size fractions.

The diversity estimator Chao1 (Tab. 1) and the rarefaction curves (Fig. 2) predict a small diversity for archaea in Mainz air (Chao1; 64 and 41 for coarse and fine, respectively). On the other hand, the relative abundances of the OTUs and the diversity calculated by Shannon (H) or Simpson (D) (Tab. 1) is slightly higher for the fine particle fraction. This might be because of the small sequence number, but is surely driven by the relative dominance of Nitrososphaeraceae sequences in the coarse particulate

matter (Fig. 1).

Most results of this study are in agreement with the previous Sanger-sequencing based study of by Fröhlich-Nowoisky et al. (2014) who analyzed 47 air filter pairs including the 20 filter pairs we focussed on in this study. However, in Fröhlich-Nowoisky et al., (2014), only a limited number of



clones were sequenced resulting in a total of 435 sequences, as compared to 2,341 sequences obtained from the current study (Tab.1). Fröhlich-Nowoisky et al., (2014) concluded that archaea occur far more often in coarse than in fine particulate matter as archaeal DNA could only be detected on 21% of the fine particle filters which is consistent with the results of this study. Further consistency is the high

abundances of Group I.1.b on coarse particle samples monitored by Fröhlich-Nowoisky et al., (2014), which now can be explained by the higher relative abundance of Nitrososphaeraceae in the coarse fraction.

The main difference between the Sanger and the Illumina approach is the estimated species richness, with 137 species from Sanger estimating almost the double amount than the Illumina approach, which

estimates 63 species. This can be caused by several issues. First, a possible lack of taxonomical depth caused by the shorter sequences compared to the Sanger approach. Second, by the closed-reference based taxonomic assignment and a possible lack in the used reference dataset. And third, and most likely, by the smaller number of sequences from more filter samples used in Fröhlich-Nowoisky et al., (2014).

As the used primers also amplified bacterial sequences, the following observation could be made: The ratio of archaea and bacteria suggests a very low proportion of airborne archaea in comparison to airborne bacteria (Fig. 3). In total, 0.07 % of the total reads could be assigned to archaea, while the rest ($5.7 * 10^6$ reads) consists of bacterial, mitochondrial, and plasmid DNA. After the sequences of mitochondria and plastids are eliminated, still the ratio of archaea to bacteria increases only to 0.1 %,

which is widely different to the ratios discovered in soil and marine environment.

This extremely low ratio is an interesting phenomenon as in most possible emission sources the proportion of archaea is higher than in air.

Several studies, focusing on airborne bacteria and archaea found that archaeal DNA in air is extremely low (Cao et al., 2014; Yooseph et al., 2013). Cao et al. found a proportion of 0.8 % of archaea when

compared to bacteria in PM10 and PM2.5 using Illumina HiSeq data (2014). Yooseph et al., (2013), who analysed the urban prokaryotic metagenome of New York, on a multistep approach based on taxonomic classifications for their peptides and assigned to the different organism groups, found that 0.48 % of their sequences were archaeal, with roughly 80 % Euryarchaeota and 20 %



Crenarchaeota/Thaumarchaeota. Both studies therefore agree with the 0.1 % archaeal sequences found in Mainz air.

Next to comparisons of species diversity and composition, also the ratio of bacteria to archaea might be an indicator of the possible emission sources, as the aerosolization process likely aerosolizes all
microorganisms at the emission source equally well. We therefore compared the ratio here detected, with ratios of possible emission sources like upper soil, ocean, and the phyllosphere (Fig. 3).

We found that compared to soil, the microbial habitat, that is often discussed as the primarily emission source, differs strongly from our and other air studies. Although, in aerated soils the archaeal abundance increases with depth (Kemnitz et al., 2007), the proportion known for surface soil is still much higher
than the proportions in air. Thus, soil alone seems an unlikely emission source. Also in sea water archaea seem to play an important role, as their abundance increases with depth up to 39 % (Karner et al., 2001). Thus, emissions from water is also unlikely the only source for airborne archaea. As Mainz is not close to oceans the only larger emission surface for water might be the river Rhine thus water, as a primary emission source in the studied area, is very likely.

In a review by Vorholt (2012) it is convincingly shown that the abundance of archaea in the phyllosphere is less than 1 % of the total microorganism load (Fig. 4), which is similar to the 0.1 % we found. With a total area of $10^9$ km$^2$ of upper and smaller leaf surface, the phyllosphere surface habitat is approximately twice the size of the land surface and is supposed to comprise up to $10^{26}$ cells worldwide (Vorholt, 2012), therefore it could present a significant emission source (Woodward and Lomas, 2004)
in the studied area and the ratio of archaea and bacteria on the phyllosphere is very close to the ratio detected in the air filter samples. Thus, the phyllosphere might be the local primary emission source.

The situation might, however, differ for individual groups found in the air filter samples, namely the Nitrososphaera family. This family includes typical soil microorganism, which points to soil as primary emission source, as these archaea were detected in high numbers. The presence of this family in the air
might be on the one hand caused by the diversity of the phyllosphere. Especially for annual plants the microorganism diversity of the phyllosphere is primarily driven by the surrounding soil and the soil microbiome sampling site (Knief et al., 2010). On the other hand, the explanation especially for the findings in the coarse fraction is that larger soil particles carry many typical soil archaea. Thus, based on



the proportions of bacteria and archaea, the most likely interpretation is, that the microbiome detected in Mainz air is primarily originating from the phyllosphere and complemented by small soil particles, which add very typical soil archaea to a great extent. Unfortunately, there is a lack of literature answering the question which archaea are typical for the phyllosphere, thus the identification of the

emission source based on the composition cannot be answered in detail.

Based on the identified genera, however, the phyllosphere and the soil can both be the primary emission source. But as the microbiome of the soil drives the one of the phyllosphere comparing taxonomy alone will anyway not lead to a final answer.

### 3.2   Seasonal dynamics

To better understand the seasonal dynamics of archaea in the atmosphere the availability of emission sources over different seasons per year can be analysed. Within this study, from the 2,341 archaeal sequences 168 could be assigned to Euryarchaeota and were studied for their seasonal behaviour. By their relative frequencies of occurrence (RFO) Thaumarchaeota are present all over the year, whereas Euryarchaeota are less abundant and their RFO values show seasonal peaks in spring and fall (Fig. 4).

Although this seasonal behavior of the Euryarchaeota agrees with the findings observed in Fröhlich-Nowoisky et al., (2014), the relative occurrence over the year seems to be larger than believed. Fröhlich-Nowoisky et al., (2014) suggested the nearby river Rhine as a potential permanent source for Methanomicrobiales and Thermoplasmatales as they are known to be present in river water throughout the year (Auguet et al., 2009; Cao et al., 2013). The specific RFO values of these orders as presented in

Fig. 5 draws, however, a slightly different picture: Methanomicrobia were observed in every season with RFO values around 40 %, thus the Rhine could contribute continuously to the aerosolized Methanomicrobia. However, the Thermoplasmata group was exclusively found in summer and fall samples, arguing against an emission from an omnipresent source like the Rhine.

Alternatively to the Rhine, potential emission sources for several euryarchaeotic groups -especially in

agricultural areas as around Mainz - are biogas substrates and life stock fertilization methods (Fröhlich-Nowoisky et al., 2014). Fig. 5 shows that Methanomicrobia and Methanobacteria both have their highest relative RFO during fall and another increase during the springs in 2006 and 2007. This



supports the hypothesis, of life-stock manure being a possible emission source, as both classes are commonly known to be present in the microbiome of live-stock and the typical times for fertilization of fields with manure is in spring and fall (Nicol et al., 2003; Radl et al., 2007). Like all methanogen groups, they have been reported in biogas reactors, too (Jaenicke et al., 2011). For the Thermoplasmata

the peaks in summer and fall might be linked to the usage of biogas reactor substrates, which are also applied to agricultural fields as fertilizer. The different RFO values of Thermoplasmata and other Euryarchaeota likely might be caused by their sensitive reaction to temperature and especially to pH, which only allows their survival in moderate or high temperatures and in low pH level environments.

The hypothesis that aerosolized archaea are linked to agricultural activities is also supported by the

seasonal variation of RFO of the order of the Nitrososphaerales within the Thaumarchaeota that is also present in the Euryarchaeota classes as discussed. Nitrososphaerales were found in agricultural soil samples close to the sampling area of our study (Ochsenreiter et al., 2003; Zhalnina et al., 2013), and thus can be considered a typical soil microorganism for agricultural soils.

## 4 Conclusion

This study gains a deeper insight into the diversity of airborne archaea. The overall abundance of archaea in the atmosphere compared to bacteria is very low, comparable to the ratio found for the phyllosphere. We found the Nitrososphaeraceae family out of the I.1.b group of Crenarchaeota to be the major archaeal family in course particulate matter. The groups Cenarchaeaceae, Nitrososphaeraceae, Methanosarcinales, Thermoplasmata and the genus *Nitrosopumilus* could be observed within the fine

particulate matter.

The observed seasonal dynamics for the dominating Euryarchaeota classes and Crenarchaeota orders, which peak in summer and fall, might be a result of agriculture in the surrounding area. So anthropogenic activities like fertilization with livestock manure or substrates of biogas reactors might influence the diversity of airborne archaea as their occurrence is increased during the main fertilization

seasons.

This combination of findings provides support for the conceptual premise that the occurrence of archaea in air might be driven by the microbiota of the phyllosphere but the influence of livestock manure gains

an edge over the phyllosphere through the fertilization seasons. Additionally groups emitted with soil as carrier particles seem to have a major influence on the community composition. For a further understanding of the dependencies of airborne microorganisms on their sources, future studies should additionally explore possible source habitats to gain as complete pictures as possible.

5    We conclude that the understanding of the seasonality, diversity, and composition of airborne archaea as one very small fraction within the bioaerosols is an important contribution to understand the patterns driving the whole atmospheric microbiome.

## 5    Data availability

The post-library-split sequence dataset will be made available from the edmond digital repository
10   http://edmond.mpdl.mpg.de/imeji/.

## Competing interests

The authors declare that they have no conflict of interest.

## Acknowledgements

The authors are grateful to M. O. Andreae, T. Andreae, J. Cimbal, R. Conrad, P. E. Galand, M. Grant,
15   M. Klose, I. Müller-Germann, C. Osterhof, C. Pio, C. Ruzene Nespoli, B. Schmitt, and H. Yang for technical assistance, as well as H. Paulsen, R. Stantscheff and Z. Jia for helpful discussions. We also thank the Mainz Bioaerosol Laboratory (MBAL) for support. This work was supported by the Max Planck Society.

J. Wehking acknowledges the PhD grant of the Konrad-Adenauer-Stiftung.

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

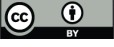



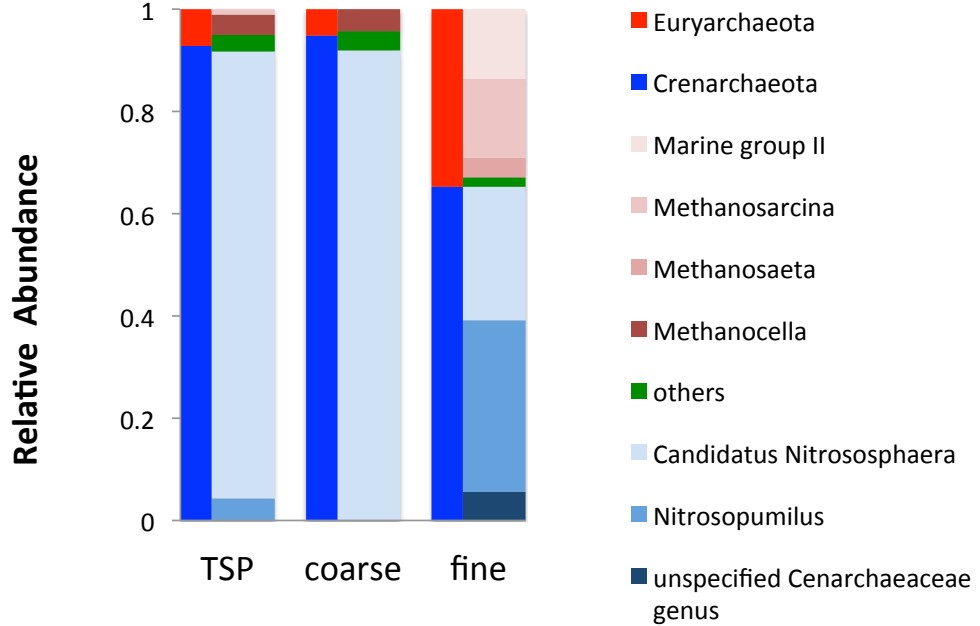

**Figure 1: Archaeal community composition for total suspended, coarse, and fine particulate matter on the level of phyla (red/blue) and genera (pastel colors).**



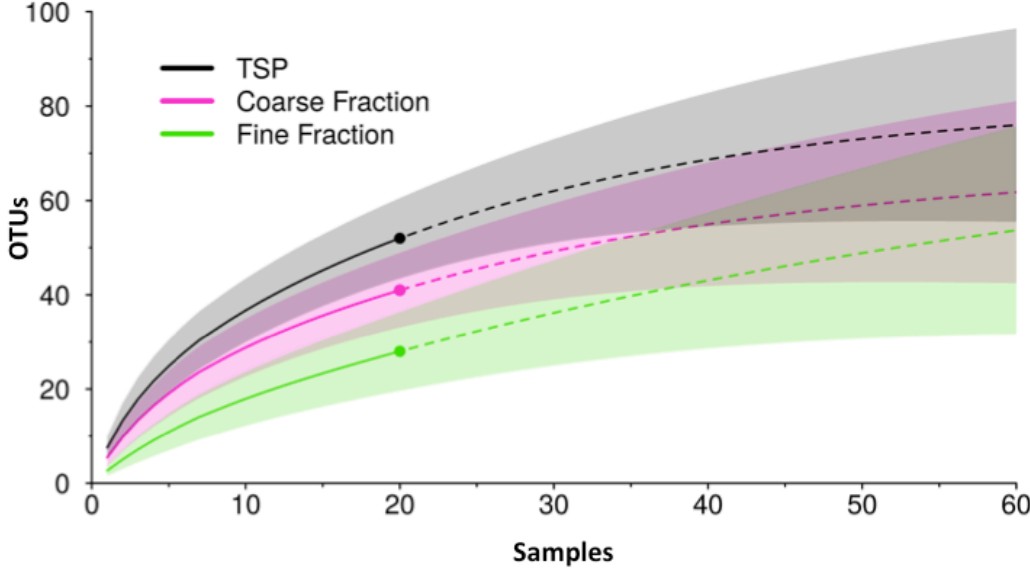

**Figure 2: Rarefaction curve of species richness. The solid curves represent the interpolated number of OTUs as a function against the number of samples. The dashed lines are according extrapolations and the dot marks the sample size of this study. The colored areas represent the 95 confidence intervals.**





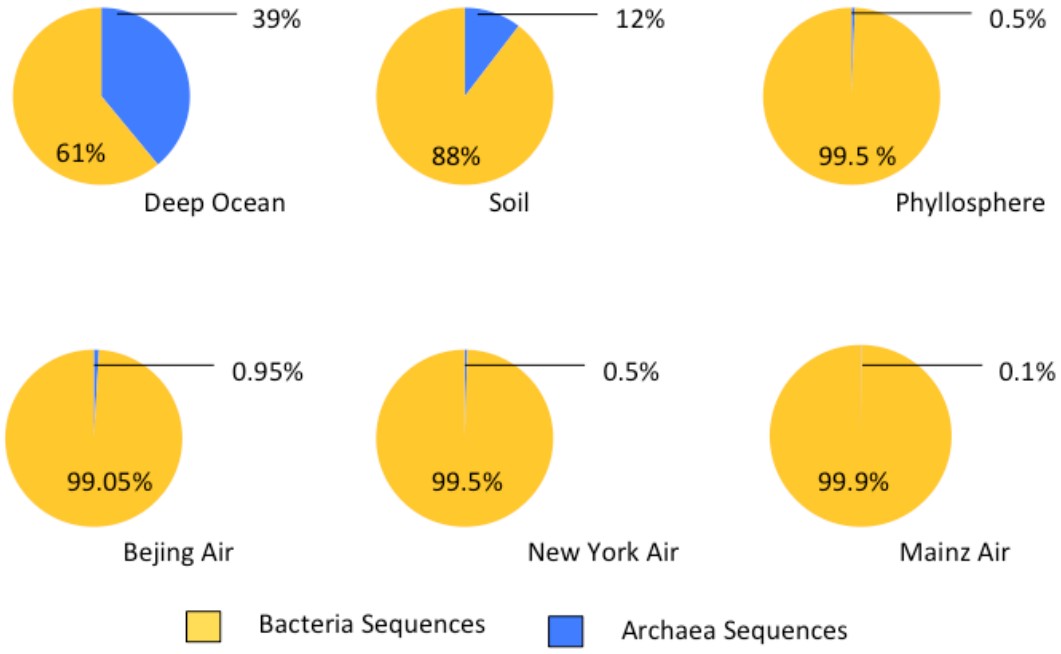

**Figure 3: Relative proportions of archaeal (blue) and bacterial (yellow) sequences detected in environmental samples. Proportions for soil are based on Kemnitz et al., (2007), for the deep ocean on Karner et al., (2001), and for the phyllosphere on Delmotte et al., (2009) and Knief et al., (2012). The proportions of the Mainz air are based on this study. The data for the New York air are**
5    **published in Yooseph et al., (2013) and the data of Bejing are based on Cao et al., (2014).**



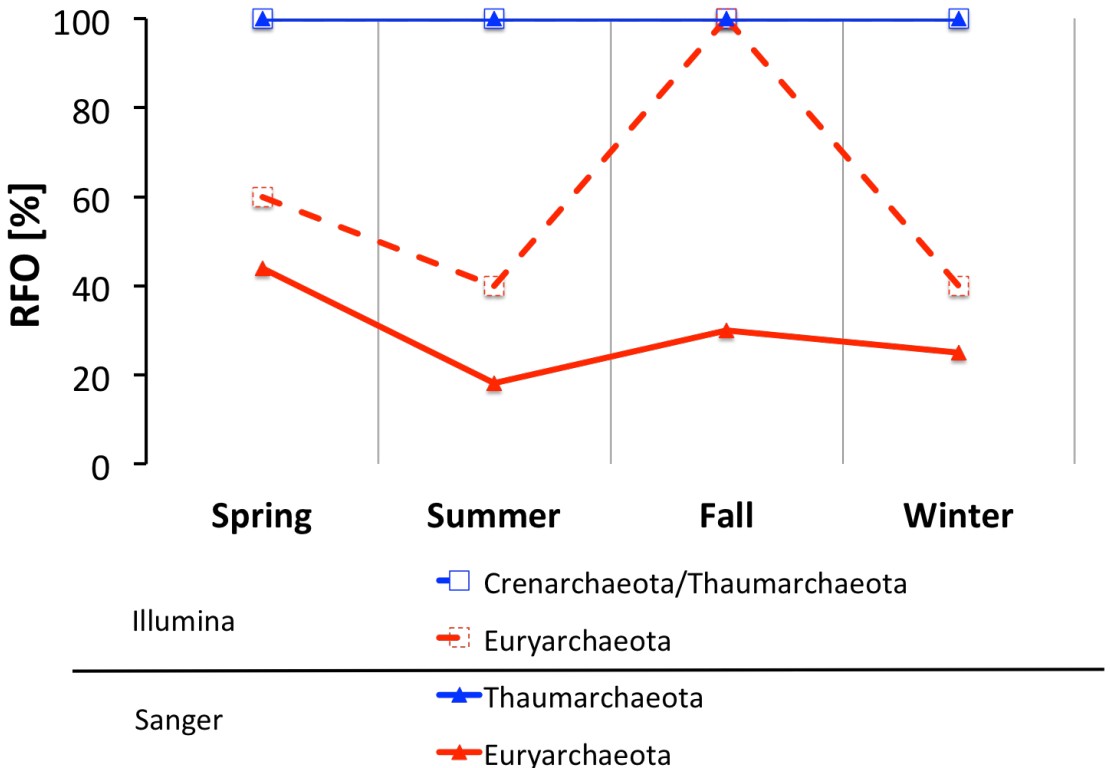

**Figure 4: Seasonal variation in the relative frequency of occurrence of airborne archaea on phylum level. The relative frequency of occurrence – the proportion of samples in which these taxa were detected - is given for both phyla, i.e., Thaumarchaeota and Euryarchaeota. The graph based on Sanger sequencing represents the data published in Fröhlich-Nowoisky et al., (2014), whereas the remaining data comprises the results of this study.**



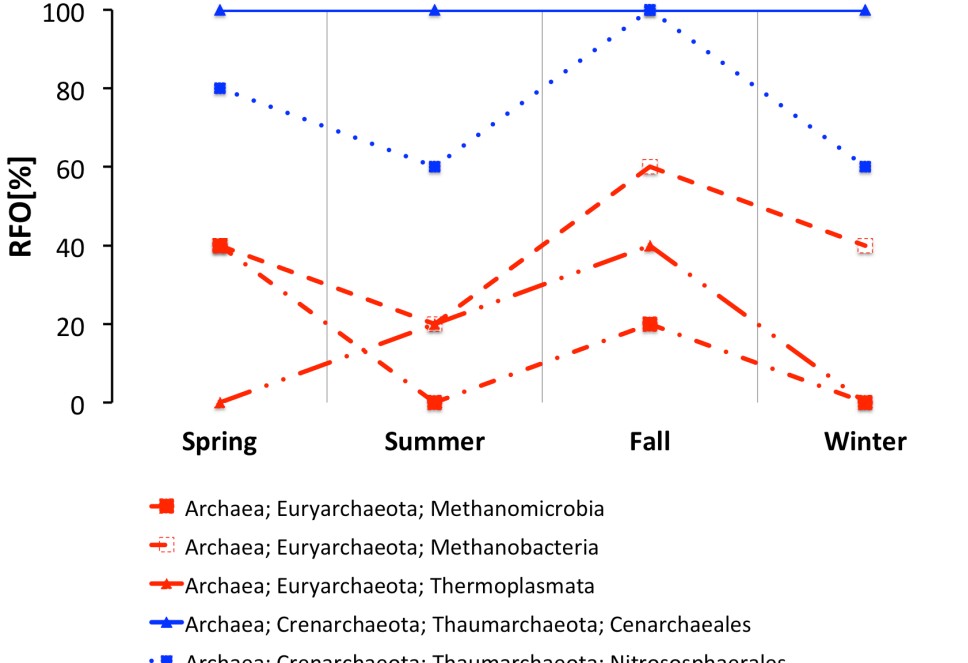

- ■ Archaea; Euryarchaeota; Methanomicrobia
- □ Archaea; Euryarchaeota; Methanobacteria
- ▲ Archaea; Euryarchaeota; Thermoplasmata
- ▲ Archaea; Crenarchaeota; Thaumarchaeota; Cenarchaeales
- ■ Archaea; Crenarchaeota; Thaumarchaeota; Nitrososphaerales

**Figure 5: Seasonal variation in the relative frequency of occurrence of dominating Euryarchaeota classes and Crenarchaeota orders.**

5    **Table 1. Number of sequences and indices estimating the archaeal diversity in Mainz for coarse and fine particle filter samples and total suspended particles (TSP).**

| Size Fraction | n (Samples) | Sq (Sequences) | Sq/n | OTU (Operational taxonomic unit) | S* (Chao1) | H (Shannon) | D (Simpson) |
|---|---|---|---|---|---|---|---|
| Coarse | 20 | 2180 | 109 | 41 | 64 | 3.09 | 0.83 |
| Fine | 20 | 161 | 8.1 | 28 | 41 | 3.65 | 0.88 |
| TSP | 20 | 2341 | 117.1 | 52 | 63 | 3.36 | 0.84 |
| Fröhlich-Nowoisky et al., (2014) | 47 | 435 | 9.3 | 57 | 137 | 3.32 | 0.82 |