# Peer review of "Community composition and seasonal changes of Archaea in 1"

_Biogeosciences, 2017_

## Referee Comment (RC1) · B. Hu (Referee) · 26 Feb 2018

The authors investigated the 16S-rRNA of airborne archaea with Illumina sequencing from atomspheric coarse and fine particulate matter samples and show seasonal dynamics and discuss anthropogenic influeces on viersity, composition and abundance of airborne archaea. The topic of this manuscript is welcomed, and the settings of the experiments are acceptable. However, the data presented in the manuscript, some 16S rRNA gene sequences and several physicochemical parameters, are too less to support a full-length scientific paper.

Specific comments: Page 9 line 18: "Candidatus Nitrosophaera" should be "Candidatus Nitrosophaera" Page 10 line 6: "Nitrososphaera" should be "Nitrososphaera"

---

## Referee Comment (RC2) · Anonymous Referee #2 · 27 Feb 2018

Interesting study and most of my comments are minor. The cited Smith paper should be cited as they identified archaea in aerosols that had crossed the Pacific. The manuscript does need a careful review for english errors. Mouse over the notes on the pdf to see suggestions on these but there are others

Please also note the supplement to this comment:
https://www.biogeosciences-discuss.net/bg-2017-514/bg-2017-514-RC2-supplement.pdf
* * *
[Figure]

**Supplement:**

[revised manuscript text omitted]

---

## Referee Comment (RC3) · Anonymous Referee #3 · 18 Apr 2018

In the present paper, Wehking et al. reported investigation of the airborne archaeal community from atmospheric coarse and fine particulate matter samples based on Illumina sequencing of 16S rRNA gene, showing its diversity, composition and abundance, discussing possible emission sources. I think it is an interest and worthwhile topic to be published.

Specific comments: Page 9 line 18: Page 5 line 23: "For the PCR amplifications the 515f/806r primer set described in Caporaso et al., (2011) proved to be most suitable." Please show the sequence of 515f/806r primer set and the evidence why to choose this primer to evaluate airborne archaeal community. Wording and gramma should be improved.

---

## Author Comment (AC1) · 3 May 2018

RC#1: The authors investigated the 16S-rRNA of airborne archaea with Illumina sequencing from atomospheric coarse and fine particulate matter samples and show seasonal dy- namics and discuss anthropogenic influseces on viersity, composition and abundance of airborne archaea. The topic of this manuscript is welcomed, and the settings of the experiments are acceptable. However, the data presented in the manuscript, some 16S rRNA gene sequences and several physicochemical parameters, are too less to support a full-length scientific paper. Specific comments: Page 9 line 18: "Candidatus Nitrosophaera" should be "Candida- tus Nitrosophaera" Page 10 line 6: "Nitrososphaera" should be "Nitrososphaera" Interactive comment on Biogeosciences Discuss., https://doi.org/10.5194/bg-2017-514, 2017

AC: We thank the referee for reviewing our study and will follow all specific comments for improvement as detailed below. With regard to the amount and relevance of the presented data, however, we do not agree with the reviewer's assessment. In an earlier article that underwent peer review, public discussion and publication in BG, we had presented, a Sanger sequencing data set with 435 Archaea sequences. In the present study, we present, analyze and discuss an Illumina sequencing data set with 2,341 sequences. To the best of our knowledge, this is the largest published dataset on airborne Archaea, and we are confident that the results and conclusions of our study are valid and merit publication in a regular research article.

Specific comments - Page 9 line 18 and Page 10 line 6: We agree and italicize both names.

---

## Author Comment (AC3) · 3 May 2018

In the present paper, Wehking et al. reported investigation of the airborne archaeal community from atomspheric coarse and fine particulate matter samples based on Illumina sequencing of 16S rRNA gene, showing its diversity, composition and abundance, discussing possible emission sources. I think it is an interest and worthwhile topic to be published. Specific comments: Page 9 line 18: Page 5 line 23: "For the PCR amplifications the 515f/806r primer set described in Caporaso et al., (2011) proved to be most suitable." Please show the sequence of 515f/806r primer set and the evidence why to choosethis primer to evaluate airborne archaeal community. Wording and gramma should beimproved.

[Figure]

AC: We thank the referee for the review and positive assessment of our manuscript, and we follow the specific comments and suggestions for improvement as detailed below.

Specific comments: Page 5 line 23: Please show the sequence of 515f/806r primer set and the evidence why to choose this primer to evaluate airborne archaeal community. Wording and gramma should be improved Current: For the PCR amplifications the 515f/806r primer set described in Caporaso et al., (2011) proved to be most suitable.

AC: For the support of the used primer set we suggest the publication of "Bates et. al. 2011" "We have demonstrated in silico that this primer set should amplify 16S rRNA genes from a broad range of archaeal and bacterial groups with few biases or excluded taxa (see Supplementary Figures S1 and S2)."

We add the Primer sequence as well as a Bates et. al. (2011) citation to the sentence: For the PCR amplifications the 515f/806r primer set (Fwd:GTGCCAGCMGCCGCGGTAA; Rev:GGACTACHVGGGTWTCTAA) described in Caporaso et al., (2011) proved to be suitable, as shown by Bates et. al. (2011).

---

## Author Response (AR1)

Dear Dr. Denise M. Akob,

We thank you for your positive evaluation of our manuscript. We would also thank the referees and are grateful for all the comments which were very helpful for improving our manuscript.

We revised the manuscript according to the referee-comments (marked in blue) and additionally improved grammar and wording (marked in red). We here submit (1) the point-by-point response and (2) the revised version of our manuscript with all changes marked as track changes.

Yours sincerely,

Jörn Wehking

B. Hu Referee #1

The authors investigated the 16S-rRNA of airborne archaea with Illumina sequencing from atomspheric coarse and fine particulate matter samples and show seasonal dynamics and discuss anthropogenic influseces on viersity, composition and abundance of airborne archaea. The topic of this manuscript is welcomed, and the settings of the experiments are acceptable. However, the data presented in the manuscript, some

16S rRNA gene sequences and several physicochemical parameters, are too less to support a full-length scientific paper.

Specific comments: Page 9 line 18: "Candidatus Nitrosophaera" should be "Candidatus Nitrosophaera" Page 10 line 6: "Nitrososphaera" should be "Nitrososphaera"

We thank the referee for reviewing our study and will follow all specific comments for improvement as detailed below. With regard to the amount and relevance of the presented data, however, we do not agree with the reviewer's assessment. In an earlier article that underwent peer review, public discussion and publication in BG, we had presented, a Sanger sequencing data set with 435 Archaea sequences. In the present study, we present, analyze and discuss an

Illumina sequencing data set with 2,341 sequences. To the best of our knowledge, this is the largest published dataset on airborne Archaea, and we are confident that the results and conclusions of our study are valid and merit publication in a regular research article.

Specific comments - Page 9 line 18 and Page 10 line 6:

We agree and italicize both names.

Anonymous Referee #2

Interesting study and most of my comments are minor. The cited Smith paper should be cited as they identified archaea in aerosols that had crossed the Pacific. The manuscript does need a careful review for english errors. Mouse over the notes on the pdf to see suggestions on these but there are others

Please also note the supplement to this comment:

https://www.biogeosciences-discuss.net/bg-2017-514/bg-2017-514-RC2- supplement.pdf

We thank the referee for the review and positive assessment of our manuscript, and we are grateful for the detailed comments which are very helpful for improving the manuscript. The specific comments and recommended changes concerning grammar and language have been implemented as detailed below.

Specific comment: Page 3 line 11 and Page 11 line 24:

Anonymous Referee #2:

"The cited Smith paper should be cited as they identified archaea in aerosols that had crossed the

Pacific."

We gladly include the additional suggested literature -Smith, D., J. Timonen, D. Jaffe, D.

Griffin, M. Birmele and M. Roberts. 2013. Intercontinental dispersal of bacteria and archaea by transpacific winds. Applied and Environmental Microbiology. 79(4):1134-1139

Specific comment: Page 5 line 10:

Anonymous Referee #2:

"this isn't clear….it reads like you overlaid the fine filter with the more coarse one but two different flow rates are presented…..03 and.27……if they were not stacked then why was the finer filter essentially free of coarse particules"

Current:

The particles with an aerodynamic diameter larger than ~3 µm and 10 % of the fine particles were collected on one glass fiber filter (~0.03 m$^3$ min-1) representing the coarse fraction. The fine particles from the same air mass were collected on the corresponding second glass fiber filter (~0.27 m$^3$ min-1) which was essentially free from coarse particles (Solomon et al., 1983)

We change the sentence as follows to clarify the fact, that the particles are split by their aerodynamic diameter into size fractions by means of a virtual impactor and not through filter pore sizes, which is not clear in the current version:

"The particles were split according to their aerodynamic diameter by means of a virtual impactor. Particles with an aerodynamic diameter larger than the nominal cut-off of ~3 µm and due to the sampling device additional 10 % of the fine particles were sampled in line with the inlet on one glass fiber filter (flowrate: ~0.03 m$^3$ min$^{-1}$) representing the coarse fraction. The fine particles were collected on a second glass fiber filter perpendicular to the inlet  (~0.27 m$^3$ min$^{-1}$) which was essentially free from coarse particles (Solomon et al., 1983)."

Specific comment: Page 5 line 13:

Anonymous Referee #2:

"collecting particles…."

Current:

Except for filter pairs MZ 11 (24 h) and MZ 15 (5 d), all filter pairs were collecting air over a 7 day period (Table S1).

We change the sentence as mentioned to:

Except for filter pairs MZ 11 (24 h) and MZ 15 (5 d), all filter pairs were collecting particles over a 7 day period (Table S1).

Specific comment: Page 9 line 18 – 20:

Anonymous Referee #2:

"rewrite"

Current:

As on coarse particle filters many more sequences could be analysed compared to the fine particle filters, analysis of the total suspended particles (TSP) resemble the results of the coarse particles (Fig. 1).

We change the sentence as follows:

Due to the much higher number of sequences isolated from the coarse particle fraction in comparison to the fine fraction the TSP composition resembles that of the coarse particle fraction (Fig. 1).

Specific comment: Page 12 line 5 - 6:

Anonymous Referee #2:

"detected here, with ratios of possible emission sources like soils, surface waters and the phyllosphere"

Current:

We therefore compared the ratio here detected, with ratios of possible emission sources like upper soil, ocean, and phyllosphere.

We change the sentence as follows:

We therefore compared the detected ratios with ratios of possible emission sources like soils, surface water and the phyllosphere reported in literature.

Specific comment: Page 13 line 15 –16:

Anonymous Referee #2:

Proposed changes.

Current:

Although this seasonal behavior of the Euyarchaeota agrees with the findings observed in

Fröhlich-Nowoisky et. al, (2014), the relative occurrence over the year seems to be larger than believed.

We change the sentence as follows:

Although the seasonal increasing or decreasing trends of the RFO values over the year are
similar to Fröhlich-Nowoisky et. al, (2014) overall, they are higher.
Specific comment: Page 13 line 19 –20:
Anonymous Referee #2:
Proposed changes.
Current:
The specific RFO values of these orders as presented in Fig. 5 draws, however, a slightly
different picture:
We agree and we change the sentence as follows:
The RFO values of the orders shown in Fig. 5 present a slightly different picture:

Anonymous Referee #3

In the present paper, Wehking et al. reported investigation of the airborne archaeal community from atmospheric coarse and fine particulate matter samples based on Illumina sequencing of

16S rRNA gene, showing its diversity, composition and abundance, discussing possible emission sources. I think it is an interest and worthwhile topic to be published.

Specific comments: Page 9 line 18: Page 5 line 23: "For the PCR amplifications the 515f/806r primer set described in Caporaso et al., (2011) proved to be most suitable."

Please show the sequence of 515f/806r primer set and the evidence why to choosethis primer to evaluate airborne archaeal community. Wording and gramma should beimproved.

We thank the referee for the review and positive assessment of our manuscript, and we follow the specific comments and suggestions for improvement as detailed below.

Specific comments: Page 5 line 23:

Please show the sequence of 515f/806r primer set and the evidence why to choose this primer to evaluate airborne archaeal community. Wording and gramma should be improved

Current:

For the PCR amplifications the 515f/806r primer set described in Caporaso et al., (2011) proved to be most suitable.

For the support of the used primer set we suggest the publication of "Bates et. al. 2011"

"We have demonstrated in silico that this primer set should amplify 16S rRNA genes from a broad range of archaeal and bacterial groups with few biases or excluded taxa (see

Supplementary Figures S1 and S2)." We add the Primer sequence as well as a Bates et. al. (2011)

citation to the sentence:

[revised manuscript text omitted]

---

## Author Response (AR2)

Dear Dr. Akob,

We are very grateful for all the comments, which were very helpful and significantly improve the manuscript.

We revised the manuscript according to your comments. We submit (1) the point-by-point response and (2) the revised version of our manuscript with all changes marked with track changes.

Yours sincerely,

Jörn Wehking

1. I think that Archaea should be capitalized throughout the paper since you are referring to a Domain of life. Same with Bacteria if you are referring to the Domain and not to all single-celled organisms.
We capitalized "Archaea" as well as "Bacteria" throughout the whole manuscript when referring to the Domain.

2. The gene studied should always be written as "16S rRNA gene" and not abbreviated to "16S" or "16S rRNA". "16S rRNA" would imply that your analysis was performed on a RNA level.
We changed to "16S rRNA gene" in the entire manuscript.

3. In SI and throughout the paper use "." instead of "," for decimal places.
We replaced the „ , " with „ . " in the whole manuscript.

4. Pg. 8, L. 22: change to 16S rRNA gene.
Was changed to "16S rRNA gene".

5. Pg. 9, l. 13: you definitely need to capitalize the domain names here
We capitalized the domain names as proposed throughout the manuscript.

6. Pg. 10, l. 17: the units are not correct. It should be "gene cp kg-1" not DNA copies. In addition, you need to clarify if you mean Bacteria and Archaea gene copies. I also would not bother abbreviating copies.
We changed the units as proposed and abstain from using "cp" as abbreviation for copies.

7. Pg. 12, l. 14: add a "." To the end of the sentence.
The full stop was already set.

8. Pg. 12, l. 6-19: the sampling time frame is not clearly described in this section. "over a 7 day period isn't clear" and 24 pairs doesn't match a full year. The only way to know when samples were collected you have to go to the SI. One-2 sentences stating that samples were collected weekly over a 1 year time period by filtering for 7 days would be helpful. State that in some cases samples were not collected because …I presume weather?

We accept that the sampling procedure is not clearly described and added the information that the presented samples were chosen out of the whole year filter set in such a way that each season is represented by 5 samples represent. We changed the original text of the section:
 "Except for filter pairs MZ 11 (24 h) and MZ 15 (5 d), all filter pairs were collecting particles over a 7 day period (Table S1)."
to
"To get a representative dataset for the whole year, five random samples, consisting of a coarse and fine filter, were analysed for each of the four seasons of the sampling campaign. The sampling period of a single filter pair was generally 7 days except for filter pairs MZ 11 (24 h), MZ 15 (5 d) and MZ 31 (5 d; Table S1)."
We hope that the sampling time frames are clarified now. Especially in combination with the changes of the next point dealing with Table S1.

9. Table S1 also has >24 samples but no information is provided about which filter each sample is. Add a column of what filter is presented. Also explain whether extracts from different filter types were combined. Provide details on which samples were which controls/blanks.

Table S1 gives all filter samples analyzed in the current study as well as in Fröhlich-Nowoisky et al (2014). We marked all filter samples which were sequenced with NGS technique additional in the updated version of table S1. Furthermore, we added the blank filters to table S1 to show in which time of the campaign the blanks were taken. The Blank filters have originally been listed in table S2, to clarify this we added "and S2" in the main manuscript. As no extracts were combined we did not mention that in the manuscript.

10. Pg. 12, l. 25: can you provide some more information on how DNA was extracted? A kit name would be sufficient

We added the missing information as proposed. The "MoBio PowerMag Soil DNA Isolation kit" was used.

11. Pg. 12: l. 26: I don't think this is correct. Illumina did not develop or have instruments that ran pyrosequencing chemistry. Illumina uses "sequencing by synthesis (SBS)" and according to this fact sheet the GAIIx instrument used SBS chemistry: next In addition, only Illumina protocols are listed on the EMP website. Please confirm the type of sequencing chemistry used and use correct language throughout the paper. I usually use "Illumina sequencing" or "next-generation sequencing".

We changed it to the term "next-generation-sequencing" and named it to "sequencing by synthesis" as proposed.

12. Pg. 13, l. 15: change to QIIME

Was changed as proposed.

13. Pg. 16, l. 7: change to "," for 2,342 sequences

Was changed as proposed.

14. Make it clear that the sequences recovered from TSP are just a sum of the coarse and fine samples. State this in the figure legends.

As the TSP flow is split into both fractions we added this information into the figure caption of Fig.1 and Fig.2

15. Pg. 18, l. 14: remove "of"
Was changed as proposed.

16. Pg. 18, l. 26: were the exact same primers used? If not, that is another important factor that could have contributed to differences in the results.
We added "and the usage of different primer pairs" to clarify that this might be another factor.

17. Figure 3: consider putting your data first (upper left most location) and putting "this study" under the location. This way you can have the reader focus more on your findings than previous work.
We changed the order like proposed.

18. Pg. 19, l. 15: if Yooseph et al. 2013 analyzed peptides then you can't present the data as sequences in Figure 3. Revise as needed to present the data accurately.
We corrected this with "metagenomic reads" as they used DNA sequences but used a combination of nucleotide and amino acid searches to assign taxonomy to their metagenomic reads.

19. Pg. 20, l. 20: is euryarchaeotic the correct term?
We rephrased to "groups of Euryarchaeota".

20. Pg. 20, l. 24 & 25: change to livestock
We changed "live-stock" to "livestock".

21. Figure 1: be mindful of whether this figure is legible by those that are color blind. Add "TSP" to the legend.
We optimized the figure for red-green color blindness and added (TSP) to the figure caption.

22. Figure 4: add RFO to the legend.
We added "RFO" to the figure caption.

23. Figure 5: specify in the legend that the data are from this study.
We added "within this study" to the figure caption.

24. Table 1: define the * after "S"

We defined it to "$S_{Chao1}$" to clarify.

[revised manuscript text omitted]

$\sum$ (-contaminants)                                                    2180 → 161

Table S2: Air filter blank samples analyzed for archaeal contamination. All families found on mounting blank filters comprise together 410 sequences: families were discarded from the data if present in more than 1% of all detected archaeal sequences on the mounting blanks, i.e., the Methanoregulaceae (8.54%), Methanomassiliicoccaceae (17.56%), and the Methanobacteriaceae (72.44%).

|  | MZ 23 |  | MZ 73 |  |  |  |
| --- | --- | --- | --- | --- | --- | --- |
|  | coarse | fine | coarse | fine | $\sum$ | % |
| Cenarchaeaceae | 0 | 0 | 1 | 1 | 2 | 0.49 |
| Methanobacteriaceae | 0 | 0 | 297 | 0 | 297 | 72.44 |
| Methanoregulaceae | 0 | 0 | 0 | 35 | 35 | 8.54 |
| Methanosaetaceae | 0 | 2 | 1 | 1 | 4 | 0.98 |
| Methanomassiliicoccaceae | 0 | 0 | 72 | 0 | 72 | 17.56 |

---

## Author Response (AR3)

Dear Dr. Akob,
We are very grateful for your positive assessment of our manuscript. All your comments within
the review process were very helpful and really helped to significantly improve the manuscript.
We revised the manuscript according to your technical comment. We submit (1) the point-by-point
response and (2) the revised version of our manuscript with the change within the figure caption
marked with track changes. We additionally made the sequence dataset public available and added
the doi therefore.
Yours sincerely,
Jörn Wehking
(1) The only correction needed is the legend for Figure 2. Please modify to:

[revised manuscript text omitted]